# Sodium and Salt Consumption in Latin America and the Caribbean: A Systematic-Review and Meta-Analysis of Population-Based Studies and Surveys

**DOI:** 10.3390/nu12020556

**Published:** 2020-02-20

**Authors:** Rodrigo M Carrillo-Larco, Antonio Bernabe-Ortiz

**Affiliations:** 1Department of Epidemiology and Biostatistics, School of Public Health, Imperial College London, London W2 1PG, UK; 2CRONICAS Centre of Excellence in Chronic Diseases, Universidad Peruana Cayetano Heredia, Lima, Lima 18, Peru; 3Centro de Estudios de Población, Universidad Católica los Ángeles de Chimbote (ULADECH-Católica), Chimbote 02804, Peru; 4Universidad Científica del Sur, Lima 15067, Peru

**Keywords:** sodium chloride, sodium, dietary, global health

## Abstract

Sodium/salt consumption is a risk factor for cardiovascular diseases. Although global targets to reduce salt intake have been established, current levels and trends of sodium consumption in Latin America and the Caribbean (LAC) are unknown. We conducted a systematic review and meta-analysis of population-based studies in which sodium consumption was analyzed based on urine samples (24 h samples or otherwise). The search was conducted in Medline, Embase, Global Health, Scopus and LILACS. From 2350 results, 53 were studied in detail, of which 15 reports were included, providing evidence for 18 studies. Most studies were from Brazil (7/18) and six collected 24 h urine samples. In the random effects meta-analysis, 12 studies (29,875 people) were analyzed since 2010. The pooled mean 24 h estimated sodium consumption was 4.13 g/day (10.49 g/day of salt). When only national surveys were analyzed, the pooled mean was 3.43 g/day (8.71 g/day of salt); when only community studies were analyzed the pooled mean was 4.39 g/day (11.15 g/day of salt). Studies had low risk of bias. The estimated 24 h sodium consumption is more than twice the World Health Organization recommendations since 2010. Regional organizations and governments should strengthen policies and interventions to measure and reduce sodium consumption in LAC.

## 1. Introduction

High sodium and sodium chloride (regular salt) consumption is a risk factor for raised blood pressure [1] and cardiovascular diseases, [2,3] which are leading causes of death globally as well as in Latin America and the Caribbean [4]. To reduce the burden of this risk factor, international organizations have established limits to its consumption: the American College of Cardiology/American Heart Association suggests that sodium intake should be lower than 1.5 g/day, [5] while the World Health Organization recommends 2 g/day [6]. However, whether current levels or trends in Latin American and the Caribbean countries are close to or far above these limits, is largely unknown.

Global efforts have aimed to quantify sodium consumption in all countries and territories, and estimates were made for 2010 [7]. Consequently, current trends, i.e., in the last decade, are unknown. A recent systematic review tried to update these estimates, yet they focused on national surveys in which 24 h urine samples were collected [8]. In both cases [7,8], the inclusion of studies from Latin America and the Caribbean was scarce. This limits the information available for the region and hampers the regional capacity to monitor sodium intake and meet global targets to achieve sodium/salt intake reduction. [9]

Therefore, to improve the evidence for Latin America and the Caribbean so that it can inform local policies and surveillance systems, we aimed to synthesize the available information on sodium consumption at the general population level in these countries. For this, a systematic review was conducted to provide recent population-based estimates on sodium consumption using urine samples (i.e., 24 h, 12 h, overnight or spot samples); we also highlighted research needs and opportunities. 

## 2. Methods

### 2.1. Protocol

Following the PRISMA guidelines Table 1 (Appendix A), we conducted a systematic review and meta-analysis of the scientific literature. 

### 2.2. Eligibility Criteria

We sought for population-based studies (i.e., nationally representative or community studies) following a random sampling technique; that is, we did not include studies with convenience samples or in which patients (e.g., only people with hypertension) or high-risk individuals (e.g., only smokers) were studied. Studies should have measured sodium in urine samples, regardless of laboratory method or collection time period (e.g., spot sample, overnight, 12 h or 24 h urine collection); however, we did not include studies which assessed sodium consumption based on questionnaires (e.g., 24 h dietary recall questionnaire) or following any other self-reported method. We focused on adult men and women from Latin American and the Caribbean countries; therefore, we did not include studies which sampled Latino populations in other countries, or which focused on foreigners living in Latin American and the Caribbean countries. 

### 2.3. Information Sources

We sought five search engines: Medline, Embase and Global Health through Ovid, as well as Scopus and LILACS. No language restrictions were set, and we restricted the search to studies from 1990 onwards. We made this decision because studies with urine samples were uncommon before 1990, and because we aimed to summarize recent trends rather than historic trends. The search was conducted on 20 December 2019. The search terms we used are available in Appendix A pp. 05–07.

### 2.4. Study Selection

Following the eligibility criteria, titles and abstracts were screened by two reviewers independently (RMC-L and AB-O); discrepancies were solved by consensus. After the screening phase, selected reports were studied in detail by two reviewers independently (RMC-L and AB-O); if there were discrepancies, these were solved by consensus. Finally, when two or more reports used the same data (e.g., publically available national surveys), we selected the oldest report or the one which presented the most information.

### 2.5. Data Collation

The authors developed a data extraction form including the following information: publication year, data collection year, country, study scope (nationally representative or community study), sample size, mean age, proportion of male subjects, proportion of people with hypertension, sodium assessment method (whether a formula was used such as the Kawasaki or Tanaka equations), urine sample collection (spot, overnight, 12 h or 24 h), and estimated 24 h sodium consumption. The data extraction form was not modified during data collation. Data extraction was conducted by one reviewer (RMC-L) and independently verified by another reviewer (AB-O); discrepancies were solved by consensus. The data collation form is available in Table 2 and Appendix A. 

### 2.6. Risk of Bias

Risk of bias of independent studies was conducted following the Newcastle–Ottawa quality assessment scale for cohort studies [24]. One reviewer (RMC-L) assessed the risk of bias and this process was independently verified by another reviewer (AB-O); discrepancies were solved by consensus. 

### 2.7. Synthesis of Results

Results are presented qualitatively and quantitatively. First, we summarized study characteristics as well as estimated 24 h sodium consumption, as reported in the original studies. However, for consistency, sodium consumption results were transformed from the original units (e.g., mmol or mEq) to grams per day (g/day). To translate sodium consumption into (sodium chloride) salt consumption, sodium consumption (in g/day) was multiplied by 2.54. Second, following a random effects meta-analysis approach, we estimated the pooled mean 24 h sodium consumption, yet we restricted this analysis to studies published since 2010. We made this decision to summarize current estimates, thereby providing timely results; notably, there were few results before 2010 and including these in the analysis did not substantially change the results (Appendix A). The pooled mean 24 h sodium consumption was also stratified by duration of urine sample collection (24 h versus otherwise) and by study scope (national versus community). In exploratory analysis, also including studies published since 2010, we conducted a meta-regression in which the dependent variable was 24 h sodium consumption (continuous numeric variable in g/day) and the independent predictors were study characteristics: sodium assessment method, duration of urine sample collection, mean age, proportion of men, proportion of people with hypertension, publication year, and study scope.

### 2.8. Ethics

No ethical approval was sought as this is a systematic literature review in which no human subjects were studied. The funder had no role in the study design, data collation, and results analysis or results presentation. The authors alone are responsible for the opinions in this work. All authors had access to the data, are responsible for its accuracy and approved the submitted work. 

## 3. Results

### 3.1. Study Selection

The search yielded 2350 results, and 2297 were excluded at the screening stage; thus, 53 were studied in detail. Finally, 15 reports (i.e., individual papers) were included in the systematic review providing results for 18 studies (Figure 1) [9,10,11,12,13,14,15,16,17,18,19,20,21,22,23]. Two studies were excluded from meta-analysis (quantitative synthesis of results) because numeric results were not available, i.e., they only reported proportions or figures without the exact numbers [13,18]. 

### 3.2. Study Characteristics

Brazil contributed to the review with seven studies [11,14,15,17,18,19,20], Chile with four [11,16,21,23] and Argentina with two [10,11]; whereas Barbados [9], Colombia [11], Ecuador [13], Peru [22] and Uruguay [12] contributed with one study each (Table 1). Three studies analyzed a nationally representative sample (Table 1) [9,15,21].

Six studies collected 24 h urine samples [9,12,14,19,22,23], while the others collected 12 h [17,18], overnight [10] or spot samples (Table 1) [11,15,16,20,21]. Eight studies used a formula (e.g., Kawasaki [25] or Tanaka [26]) to estimate 24 h sodium excretion (Table 1) [11,15,16,20,21]. 

### 3.3. Quantitative Synthesis of Results

The meta-analysis included 12 studies published since 2010 (29,875 people), [9,11,12,14,15,19,21,22,23] with a mean age ranging from 41.2 to 53.5 years [14,23]; the proportion of men varied between 33.7% and 48.2% [11,23]. The proportion of people with hypertension went from 17.0% to 52.5% (Table 1) [11,22]. 

None of the summarized studies reported mean estimated sodium consumption below the World Health Organization recommendations (2 g/day) [6]. The estimated 24 h sodium consumption varied from 2.66 (standard deviation: 1.64) [9] g/day to 4.89 (standard deviation: 1.48) [11] g/day (Table 2). 

The random effects meta-analysis including studies published since 2010 revealed a pooled mean estimates 24 h sodium consumption of 4.13 g/day (95% confidence interval: 3.82–4.44, I^2^: 99.7%) [9,11,12,14,15,19,21,22,23], which will be equivalent to 10.49 g/day of salt. 

When these studies were further divided, those based on a national sample exhibited a pooled mean estimated sodium consumption of 3.43 g/day (95% confidence interval: 3.12–3.75, I^2^: 99.0%) [9,15,21]; those based on a community sample reported a pooled mean of 4.39 g/day (95% confidence interval: 4.22–4.56, I^2^: 97.8%) [11,12,14,19,22,23]. The equivalent estimated salt consumption was 8.71 g/day and 11.15 g/day, respectively. 

The pooled mean estimated sodium consumption amongst studies with urine samples collected after 24 h was 3.82 g/day (95% confidence interval: 3.27–4.37, I^2^: 97.9%) [9,12,14,19,22,23]; the pooled mean estimated sodium consumption amongst studies with samples collected in a different fashion to 24 h collection was 4.43 g/day (95% confidence interval: 4.01–4.85, I^2^: 99.8%) [11,15,21]. The equivalent estimated salt consumption was 9.70 g/day and 11.25 g/day, respectively. 

We further compared estimated sodium consumption per 24 h urine samples [9,12,14,19,22,23] versus the Tanaka [15,16,21] and Kawasaki [11] formula. The pooled mean 24 h estimated sodium consumption across these three groups were: 3.82 (95% confidence interval: 3.27–4.37, I2: 97.9%), 3.78 (95% confidence interval: 3.58–3.98, I^2^: 98.0%) and 4.75 (95% confidence interval: 4.58–4.91, I^2^: 97.9%), respectively. 

Although there was a limited number of studies per year, there seemed to be an increasing trend, which has reverted recently. The mean estimated 24 h sodium consumption in the study published in 2015 was 4.06 g/day (95% confidence interval: 3.86–4.26); for studies published in 2016 the pooled estimated mean was 4.67 g/day (95% confidence interval: 4.51–4.83, I^2^: 97.5%); for studies published in 2018 the pooled mean was 3.67 g/day (95% confidence interval: 2.90–4.46, I^2^: 98.6%); finally, for those published in 2019, the pooled mean was 3.78 g/day (95% confidence interval: 3.58–3.98, I^2^: 98.0%). 

In meta-regression analysis including only studies published since 2010, one predictor was strongly associated with mean estimated 24 h sodium consumption (Table 3). Community studies were positively associated with higher mean estimates (Coef: 0.96, 95% confidence interval: 0.23; 1.68, *p*-value: 0.015). In addition, in comparison to 24 h urine samples, a meta-regression model adjusting for study scope (national or community) revealed that the Tanaka formula retrieved slightly overestimated results (Coef: 0.56, 95% confidence interval: −0.05; 1.16, *p*-value: 0.067), yet the Kawasaki formula showed a stronger association (Coef: 0.77, 95% confidence interval: 0.30–1.25, *p*-value: 0.005). 

### 3.4. Risk of Bias

All reviewed studies had a low risk of bias (Table 4 and Appendix A). 

## 4. Discusion 

### 4.1. Summary of Evidence

In this systematic review to study sodium consumption in Latin America and the Caribbean, the pooled mean estimated 24 h sodium consumption since 2010 was 4.13 g/day (10.49 g/day of salt), more than twice the World Health Organization recommendations. Interestingly, these estimates varied depending on the study scope. The pooled mean estimated 24 h sodium consumption from national samples was notoriously lower (3.43 g/day of sodium) compared to the pooled mean from community samples (4.39 g/day of sodium). We also observed some overestimation when studies relied on the Kawasaki or Tanaka formula. Although the meta-analytic results need to be interpreted in light of the heterogeneity of populations and methods, these preliminary findings could suggest that national studies are most needed, along with research to estimate 24 h sodium consumption with easier, but accurate, algorithms/formulae. 

While acknowledging the limitations of this review along with the limitations of the original reports, the results suggest that sodium consumption in several countries in Latin America and the Caribbean is above the international recommendations [5,6]. Although universal health coverage is underway and will secure treatment for people with diseases such as hypertension, population-based policies and interventions are needed to curtail sodium consumption and to improve the intake of alternative minerals (e.g., potassium chloride salt). 

### 4.2. Results in Context 

Besides the INTERSALT study conducted in the late 1980s and not included in this review [27], and the study conducted by Lamelas and colleagues in four countries in South America [11], to the best of our knowledge there are no other recent regional or multi-country efforts to quantify sodium consumption in Latin America and the Caribbean. Our systematic review starts to fill this knowledge gap providing timely evidence to guide policies, interventions and future research on sodium consumption in Latin America and the Caribbean. Our estimates could inform monitoring frameworks to assess progress in the reduction in sodium consumption across the region. 

A global pooling endeavor which provided estimates until 2010 reported a daily sodium intake in Latin America and the Caribbean raging between 2.6 and 4.1 g/day [7]. The results we provided, nonetheless, are slightly larger. This could suggest an increasing trend since 2010; however, dissimilar methodologies with the work by Powles et al [7] could also explain the differences.

We found that most recent estimates are slightly smaller than those of previous years. A similar finding was pinpointed by a recent global work [8], in which they reported that for countries with national sodium intake estimates post 2010, these were smaller than the estimates reported for 2010 [7,8]. Although largely speculative, this could suggest that policies and population-wide interventions to reduce salt consumption may be showing positive results. Among others, these include food labeling and setting limits for certain foods (e.g., bread) [28,29]. A thorough evaluation of policies to reduce sodium/salt consumption in Latin America and the Caribbean is warranted.

Due to the limited/null number of studies for several countries, it seemed impossible to compare countries or sub-regions (e.g., Andean Latin America). The results only show that any estimate provided for South America is larger than the result for the Caribbean; this is consistent with the findings by Powel and colleagues [7]. Different dietary sources could potentially explain the differences [30]; for example, meat and bakery seem to be relevant sodium sources in low- and middle-income countries [30] and we could argue that these, particularly meat, are more often consumed in South America. Gender and age distributions in the selected reports were largely similar across the countries, which may imply that these variables do not greatly explain the results.

### 4.3. Limitations 

We followed a strong methodology using five search engines, summarizing studies with reliable methods to ascertain 24 h sodium consumption. We have summarized more results for Latin America and the Caribbean than previous and most recent global reviews [7,8].

Limitations must be acknowledged as well. First, we did not search the grey literature. Although this could have potentially provided additional reports, it is highly unlikely that these would have substantially changed the results or conclusions. These would be most likely community studies, probably without urine samples. Second, we did not retrieve enough results to compute strong meta-regressions for all potential predictors. However, the findings from the meta-regression analysis are still informative and could guide new studies or recommendations for future research. Third, we retrieved very few results from Central America and the Caribbean, which highlights the urgent need for this evidence in these sub-regions. Fourth, because few studies reported the years of data collection, we conducted the analysis based on publication year. Results about trends should be interpreted in line with this potential limitation. Fifth, we acknowledge that the overall mean sodium/salt consumption based on the meta-analytical approach may not be the best metric to provide country- or regional-specific summary measures, because few studies were national samples (i.e., accounted for population weights) and many countries did not have data available. However, this summary metric aimed to report on the regional current global landscape of (high) sodium consumption as a starting point for stronger studies to build on.

A common limitation amongst the selected studies was the inconsistent methodology they followed to ascertain 24 h sodium consumption, i.e., some authors collected 24 h urine samples while others used spot samples, or they relied on different equations to compute 24 h sodium consumption. The results showed some potential differences in sodium consumption estimates based on 24 h urine samples versus estimates based on formulas; thus, for surveillance and in large national surveys, the results do not strongly support the use of any formula in comparison to 24 h urine samples. However, in order to have consistent and comparable results across countries and time, and to reflect the most up-to-date evidence, we suggest that regional organizations (e.g., PAHO/WHO) or professional bodies (e.g., cardiology associations) should update standard guidelines [25] for researchers who want to estimate sodium consumption in Latin America and the Caribbean. Similarly, regional organizations and local governments should convene efforts to move from community studies to including sodium intake assessment in national surveys (e.g., demographic and health surveys). 

### 4.4. Public Health Implications

The World Health Organization, as well as the Pan-American Health Organization, has issued the SHAKE package for salt reduction [31,32]. This document summarizes strategies for salt reduction that countries can develop and implement [32]._ENREF_18 These strategies are meant to help countries achieve the non-communicable disease monitoring framework’s fourth aim: *30% relative reduction in mean population intake of salt/sodium* [33]. This systematic review provides initial evidence to inform this goal, by reporting current mean sodium intake since 2010 and before as preliminary evidence of time trends. These results were not available, preventing a more up-to-date and comprehensive monitoring of salt consumption in the region, while also pinpointing research needs.

A 24 h urine sample is the gold standard to ascertain sodium consumption; this approach is also recommended by the Pan-American Health Organization [22,23,25]. However, collecting 24 h urine samples could be troublesome to implement in large, population-based studies and especially in national samples. There is need for simpler methods to ascertain daily sodium intake, for example those based on spot urine samples, including some correction factors [25], as well as the Tanaka [26], Kawasaki [25], and INTERSALT [34] formulas. Although some authors have claimed that these formulas may retrieve accurate estimates [35,36], there is also a large body of evidence suggesting these formulas are not perfect or may need calibration in new populations [37,38]. The findings of this review and meta-regressions may suggest that the Tanaka and Kawasaki formula may slightly overestimate 24 h sodium consumption. A methodology systematic review reached similar conclusions: it is uncertain whether alternative methods to estimate 24 h sodium consumption are valid [37]. In a similar vein, an expert’s position statement signaled that more research is needed on the validity of alternative urine samples (e.g., spot samples) to study mean sodium consumption at the population level [39]. Based on our results regarding the Tanaka and Kawasaki formulas, as well as on international evidence suggesting that methods other than 24 h urine samples may not be adequate or need re-calibration, we recommend that a thorough evaluation, and possibly a re-calibration, of these formulas, is needed to better inform Latin America and the Caribbean mean sodium consumption estimates.

In light of the global conflicting evidence and the results herein presented, and because the Pan-American Health Organization guidelines were published 10 years ago [25], this guideline should be updated to appraise whether spot urine samples could be useful to study sodium consumption in large populations in Latin America and the Caribbean. As a matter of fact, our findings did not show a strong difference between 24 h urine samples and the Tanaka formula [26]. Based on this finding, international evidence and further analysis as needed, guidelines can be updated to make it easier for countries and researchers to monitor sodium consumption in Latin America and the Caribbean. Furthermore, our findings did support that there is a difference between sodium estimates based on a national sample versus a community sample. Therefore, studying national samples could be a priority, for which friendly and inexpensive methods are much warranted. 

From a research perspective, large studies could be conducted in Latin America and the Caribbean to provide local and regional evidence about the accuracy of available formulae and, when relevant, calibrate them. Data pooling studies, for example, those herein summarized, could be useful to adapt available formulas for populations in Latin America and the Caribbean. New technologies and prediction methodologies, for example, machine learning algorithms, could be tested for this purpose. 

## 5. Conclusions 

Sodium consumption evaluation across Latin America and the Caribbean still seems to be limited. This calls for urgently strengthening research and national surveys to also study sodium consumption in a consistent and comparable fashion. Nonetheless, available evidence already suggests that 24 h our sodium consumption is twice the international recommendations. When stronger studies become available to improve regional evidence on this matter, regional organizations and governments should strengthen policies and interventions to reduce sodium consumption in Latin America and the Caribbean. 

## Figures and Tables

**Figure 1 nutrients-12-00556-f001:**
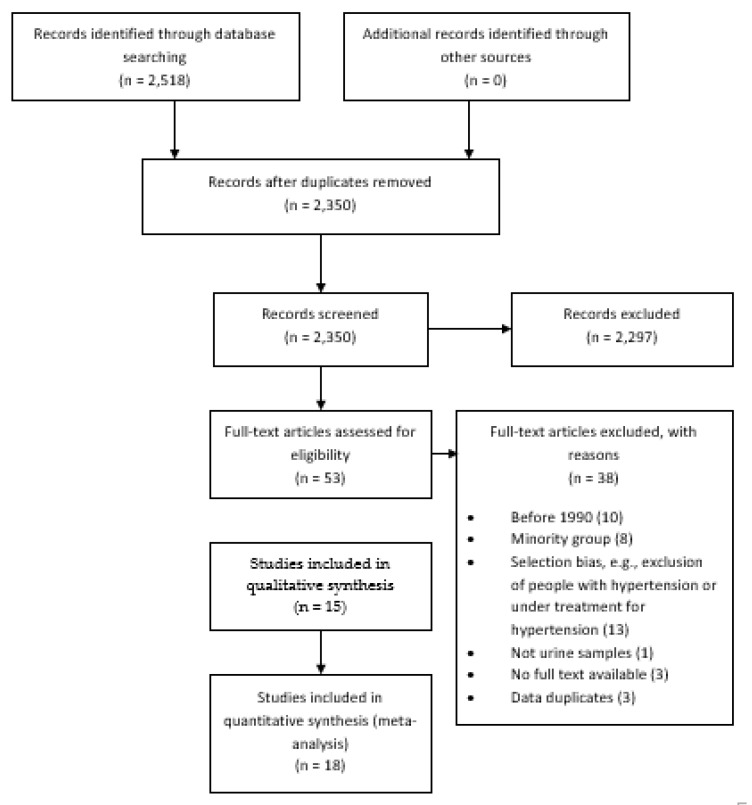
Study selection.

**Table 1 nutrients-12-00556-t001:** Study characteristics.

Author	Country	Type	Sample	Mean Age (Years)	Proportion Men (%)	Proportion HTN* (%)	Sodium Assessment	Collection Time
Carbajal, 2001 [10]	ARG	Community	1225		37.06	35.83	Results were multiplied by 3 (i.e., 24 h)	Overnight (from 23 h to 7 h)
Lamelas, 2016 [11]	ARG	Community	6529	51.10	39.00	51.60	Kawasaki formula was used to estimate 24 h sodium excretion, surrogate of daily sodium consumption	Morning fasting midstream urine sample
Lamelas, 2016 [11]	BRA	Community	5323	52.20	45.20	52.50	Kawasaki formula was used to estimate 24 h sodium excretion, surrogate of daily sodium consumption	Morning fasting midstream urine sample
Lamelas, 2016 [11]	CHL	Community	668	52.00	33.70	41.30	Kawasaki formula was used to estimate 24 h sodium excretion, surrogate of daily sodium consumption	Morning fasting midstream urine sample
Lamelas, 2016 [11]	COL	Community	4513	50.80	37.10	37.70	Kawasaki formula was used to estimate 24 h sodium excretion, surrogate of daily sodium consumption	Morning fasting midstream urine sample
Moliterno 2018 [12]	URY	Community	149	52.69	40.30	36.23	As per urine samples	Participants were instructed to collect a 24 h urine sample
Del Pozo, 1990 [13]	ECU	Community	332		47.59	~110/75		
Perin, 2018 [14]	BRA	Community	517	53.50	41.60	44.50	As per urine samples	Participants were instructed to collect a 24 h urine sample
Mill, 2019 [15]	BRA	National	8083				Tanaka formula was used to estimate the 24 h sodium excretion	Random sample (as long as the participant had gone at least 2 h without urinating
Lopez-Rodrigez, 2009 [16]	CHL	Community	48	39.00		121.5/79.5	Tanaka formula was used to estimate the 24 h sodium excretion	Random sample
Bisi, 2003 [17]	BRA	Community	1663	44.97	45.90	42.69	As per urine samples (here multiplied by 2)	12 h urine collection
Cipullo, 2010 [18]	BRA	Community	1717	55.00	48.80	44.38	As per urine samples	12 h urine collection
Rodrigues, 2015 [19]	BRA	Community	272	44.00	47.43	31.25	As per urine samples	Participants were instructed to collect a 24 h urine sample
Costa, 1990 [20]	BRA	Community	4565			11.75	Grams of NaCl per day were estimated by the antilogarithm of the average log(Na/creatinine) of spot urine samples times 1.60 (adjustment for daily creatinine) times 0.058	Random sample
Petermann-Rocha, 2019 [21]	CHL	National	2913	46.47	41.74	128.4/76.5	Tanaka formula was used to estimate the 24 h sodium excretion	Random sample
Carrillo-Larco, 2018 [22]	PER	Community	409	45.70	44.00	17.00	As per urine samples	Participants were instructed to collect a 24 h urine sample
Harris, 2018 [9]	BRB	National	364		44.23	34.10	As per urine samples	Participants were instructed to collect a 24 h urine sample
Campino, 2016 [23]	CHL	Community	135	41.2	48.15		As per urine samples	Participants were instructed to collect a 24 h urine sample

HTN: hypertension. *In three cases mean systolic/diastolic blood pressure were only reported.

**Table 2 nutrients-12-00556-t002:** Sodium consumption.

Author	Country	Sodium (g/Day)	Salt (g/Day)
Carbajal, 2001 [10]	ARG	3.02	7.68
Lamelas, 2016 [11]	ARG	4.66	11.84
Lamelas, 2016 [11]	BRA	4.57	11.61
Lamelas, 2016 [11]	CHL	4.88	12.40
Lamelas, 2016 [11]	COL	4.89	12.42
Moliterno 2018 [12]	URY	3.52	8.93
Del Pozo, 1990 [13]	ECU	4.63	11.75
Perin, 2018 [14]	BRA	4.13	10.50
Mill, 2019 [15]	BRA	3.68	9.34
Lopez-Rodrigez, 2009 [16]	CHL	4.10	10.41
Bisi, 2003 [17]	BRA	4.55	11.57
Cipullo, 2010 [18]	BRA	Urinary sodium in normotensive: <100 mEq/L = 55.1%; 100–149 mEq/L = 25.8%; ≥150 mEq/L = 19.1%. Urinary sodium in hypertensive: <100 mEq/L = 43.5%; 100–149 mEq/L = 29.8%; ≥150 mEq/L = 26.7%.	
Rodrigues, 2015 [19]	BRA	4.06	10.31
Costa, 1990 [20]	BRA	5.09	12.93
Petermann-Rocha, 2019 [21]	CHL	3.88	9.86
Carrillo-Larco, 2018 [22]	PER	4.40	11.18
Harris, 2018 [9]	BRB	2.66	6.76
Campino, 2016 [23]	CHL	4.16	10.57

Salt was estimated as sodium times 2.54.

**Table 3 nutrients-12-00556-t003:** Meta-regression analysis.

	Coef	95% confidence interval (*p*-value)
Sodium analysis (ref: as per urine samples)	*n* = 12	
Formula-based	0.60	−0.14; 1.36 (0.102)
Survey scope (ref: national)	*n* = 12	
Community	**0.96**	**0.23; 1.68 (0.015)**
Urine collection (ref: 24 h)	*n* = 12	
Different than 24 h	0.61	−0.14; 1.36 (0.102)
Proportion of men (continuous) *n* = 11	−0.06	−0.17; 0.04 (0.196)
Mean age (continuous) *n* = 10	0.03	−0.06; 0.11 (0.519)
Proportion of hypertension (continuous) *n* = 9	0.02	−0.04; 0.08 (0.476)
Publication year (continuous) *n* = 12	**−0.26**	**−0.53; −0.00 (0.049)**

Estimates in bold are those with *p* < 0.05.

**Table 4 nutrients-12-00556-t004:** Risk of bias.

Author	Country	Representativeness	Selection	Ascertainment	Assessment
Carbajal, 2001 [10]	ARG				
Lamelas, 2016 [11]	ARG				
Lamelas, 2016 [11]	BRA				
Lamelas, 2016 [11]	CHL				
Lamelas, 2016 [11]	COL				
Moliterno 2018 [12]	URY				
Del Pozo, 1990 [13]	ECU				
Perin, 2018 [14]	BRA				
Mill, 2019 [15]	BRA				
Lopez-Rodrigez, 2009 [16]	CHL				
Bisi, 2003 [17]	BRA				
Cipullo, 2010 [18]	BRA				
Rodrigues, 2015 [19]	BRA				
Costa, 1990 [20]	BRA				
Petermann-Rocha, 2019 [21]	CHL				
Carrillo-Larco, 2018 [22]	PER				
Harris, 2018 [9]	BRB				
Campino, 2016 [23]	CHL				

The more there are, the lower the risk of bias.

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
