# Peer review of "Sodium and Salt Consumption in Latin America and the Caribbean: A Systematic-Review and Meta-Analysis of Population-Based Studies and Surveys"

_nutrients, 2020, doi:10.3390/nu12020556_

Round 1
Reviewer 1 Report
The paper justifies the need for a regional overview of sodium intake for LAC and identifies currently available estimates from the region. Identifying and collating the currently available data is useful, and with a more appropriate analysis will be a significant contribution to the field. Including a search of the grey literature would be a valuable addition.
Defining a pooled mean sodium intake using this diverse data is not very meaningful, particularly in the absence of population weighting. There needs to be more discussion about the appropriateness of this analysis, and the efficacy of a regional average based on the current data. I encourage the authors to take a more qualitative approach and discuss the range of methods, differences between countries/populations within the region, and possible trends; the current statements in the discussion regarding these points overstates what can be concluded from the current data. It would also be relevant to discuss the implications of the gender and age of the study samples.
Overall the paper is clearly written. Some aspects are incomplete: table 4 is missing content; the reference list does not include the source for some of the entries, e.g. journal name; the stated number of studies included in the review and analysis is in the manuscript and in figure 1, this may be partly because the Lamelas et. al. paper included data on several populations- but it needs to be clarified.
The policy and research implications discussed are important but need to be grounded in appropriate analysis of the current data.
Reviewer 2 Report
The article is interesting and provides important and striking data regarding sodium intake, also makes a good clarification about the limitations of the study. Only there are few correction.
Thank you.

Reviewer 3 Report
In the publication, the authors raise the issue of excessive sodium / salt consumption, which is important from the point of view of public health. Controlling the supply of salt in the diet of populations of different countries seems to be important due to the currently widespread so-called civilization diseases such as hypertension.
However, the authors did not avoid a few shortcomings:
- wrong way to cite literature in the text. The numbers are written in superscript, not in brackets;
- no reference in the text to table 2;
- unscientific language ('We compare ...', 'We do not belive ....', 'our results' etc.);
- in the discussion section - summary of evidence in the first paragraph (no line numbering) is: 3.43 and 4.39 g / day salt, it should be rather 3.43 and 4.39 g / day sodium;
- the list of references should not contain entire names of magazines, but their short names
- in the list of references, when citing websites, there are no archiving dates
- the authors state that for Latin American and Caribbean data, the use of different models for assessing daily salt intake give different results and require calibration to local conditions. They do not refer to data for other regions of the world. Have similar problems been observed?
Round 2
Reviewer 3 Report
In the context of questions 1-6, I thank the authors for their comments and editorial improvement.
Regarding question 7. The authors state that their research and conclusions apply only to the region of Latin America and the Caribbean and therefore conclude that some formulas for calculating daily sodium / salt intake must be calibrated to local conditions.
This is most likely. On the other hand, in the 'Limitations' chapter, they suggest that the amount of data that is available and which the authors have analyzed may be too small, and therefore the conclusions may not be precise. Therefore, I recommend that in support of the statement:
"The findings of this review and meta-regressions may suggest that the Tanaka and Kawasaki formula may slightly overestimate 24-hour sodium consumption. A thorough evaluation, and possibly a re-calibration, of these formulas is needed to better inform Latin America and the Caribbean estimates."
the authors sholud presented whether similar or opposite conclusions as to the formulas used appear in the literature.
